

# SODU2-NET: a novel deep learning-based approach for salient object detection utilizing U-NET

Hyder Abbas[1,2], Shen Bing Ren[2], Muhammad Asim[3,4],
Syeda Iqra Hassan[5,6] and Ahmed A. Abd El-Latif[3,7]

[1] State Key Laboratory of Public Big Data, College of Computer Science and Technology, Institute for Artificial Intelligence, Guizhou University, Guiyang, Guizhou, China
[2] School of Computer Science and Engineering, Central South University, Changsha, China
[3] EIAS Data Science and Blockchain Laboratory, College of Computer and Information Sciences, Prince Sultan University, Riyadh, Saudi Arabia
[4] School of Computer Science and Technology, Guangdong University of Technology, Guangzhou, China
[5] Department of Electrical and Electronic Engineering, British Malaysian Institute, Universiti of Kuala Lumpur, Kuala Lumpur, Malaysia
[6] Software Engineering Department, Sir Syed University of Engineering and Technology, Karachi, Pakistan
[7] Department of Mathematics and Computer Science, Faculty of Science, Menoufia University, Shebin El-Koom, Egypt

Corresponding authors
Muhammad Asim,
masim@psu.edu.cn
Syeda Iqra Hassan,
iqra.hassan@ssuet.edu.pk

## ABSTRACT

Detecting and segmenting salient objects from natural scenes, often referred to as salient object detection, has attracted great interest in computer vision. To address this challenge posed by complex backgrounds in salient object detection is crucial for advancing the field. This article proposes a novel deep learning-based architecture called SODU2-NET (Salient object detection U2-Net) for salient object detection that utilizes the U-NET base structure. This model addresses a gap in previous work that focused primarily on complex backgrounds by employing a densely supervised encoder-decoder network. The proposed SODU2-NET employs sophisticated background subtraction techniques and utilizes advanced deep learning architectures that can discern relevant foreground information when dealing with complex backgrounds. Firstly, an enriched encoder block with full feature fusion (FFF) with atrous spatial pyramid pooling (ASPP) varying dilation rates to efficiently capture multi-scale contextual information, improving salient object detection in complex backgrounds and reducing the loss of information during down-sampling. Secondly the block includes an attention module that refines the decoder, is constructed to enhances the detection of salient objects in complex backgrounds by selectively focusing attention on relevant features. This allows the model to reconstruct detailed and contextually relevant information, which is essential to determining salient objects accurately. Finally, the architecture has been improved by adding a residual block at the encoder end, which is responsible for both saliency prediction and map refinement. The proposed network is designed to learn the transformation between input images and ground truth, enabling accurate segmentation of salient object regions with clear borders and accurate prediction of fine structures. SODU2-NET is demonstrated to have superior performance in five public datasets, including DUTS, SOD, DUT OMRON, HKU-IS, PASCAL-S, and a new real world dataset, the Changsha dataset. Based on a comparative assessment of the model FCN, Squeeze-net, Deep Lab, Mask R-CNN the proposed SODU2-NET is found and achieve an

improvement of precision (6%), recall (5%) and accuracy (3%). Overall, approach shows promise for improving the accuracy and efficiency of salient object detection in a variety of settings.

## INTRODUCTION

In the era of cutting-edge research, computer vision stands as an extraordinary endeavor, akin to bestowing artificial intelligence with the gift of sight (*Hazarika et al., 2022*). This interdisciplinary field amalgamates advanced algorithms, machine learning techniques, and sophisticated imaging technologies to endow machines with the ability to interpret, analyze, and comprehend visual data (*Zhang et al., 2013*). Through the intricate interplay of convolutional neural networks, feature extraction, and pattern recognition, computer vision unveils a new dimension of understanding the visual world. As researchers delve deeper into refining these intricate mechanisms, the boundaries of what can be achieved through the fusion of artificial intelligence and visual perception continue to expand, promising transformative impacts across industries and applications (*Boulila et al., 2022*).

Saliency object detection is a computer vision technique that aims to identify the most "salient" or eye-catching objects in an image (*Borji et al., 2019*). Saliency detection algorithms analyze the image's content and determine which regions or objects stand out from the background based on various visual cues such as color, texture, shape, and motion (*Jing et al., 2014*). These regions or objects are typically considered the most important or relevant in the image and are identified using saliency object detection algorithms. The term "saliency" comes from the word "salient," which means "conspicuous" or "noticeable." By identifying salient objects, computer vision systems can prioritize processing resources and focus attention on the most relevant parts of an image. The output of a saliency object detection model is a map that highlights the most salient regions, which can be useful for various computer vision tasks such as object recognition (*Zou et al., 2023*), image segmentation (*Tajbakhsh et al., 2021*), and attention-based visual recognition (*Yuan et al., 2022*).

Computer vision serves as the guiding light, unraveling intricate details and revealing the concealed essence of salient object detection within visual landscapes. Image saliency detection operates by emulating the human visual attention mechanism through computer algorithms, thus creating a saliency detection model. This process involves simulating the way humans perceive visual information, enabling the computer to pinpoint areas of significance within an image. Experiments have shown that people first pay attention to the parts of the image, thereby effectively extracting image features as prominent objects. Extracting image features using trained convolutional neural networks has become the most common and effective technique. However, with the continuous expansion of image datasets and the gradual increase in scene complexity, traditional saliency detection has

proven difficult to meet the demands of scholars. It is difficult to capture semantic information in many computer vision tasks, with significant limitations. Due to the continuous complexity of the image, existing depth significant object detection models still have drawbacks. Image saliency detection research faces several key challenges:

1) Lack of generalization: Ensuring models work well across diverse datasets and real-world scenarios.
2) Complex scenes: Accurately identifying multiple salient objects amidst clutter and occlusions.
3) Handling object interactions: Recognizing and differentiating interacting salient objects.
4) Weakly supervised learning: Developing methods that require minimal annotation efforts.

Addressing these challenges will advance image saliency detection and benefit broader computer vision applications. Therefore, in order to solve the existing problems, further research on significance detection is of great significance.

This study introduces a deep learning (DL)-based model approach called Salient object detection U2-Net (SODU2-NET) to autonomously detect noteworthy objects within images. The U-Net, a convolutional neural network (CNN) variant, is modified for image segmentation purposes (*Aboelenein et al., 2020*). Comprising both a contracting and an expansive path, the U-Net architecture represents a refined rendition of the conventional encoder-decoder network. To enhance its capabilities, the standard U-Net design is enhanced by the incorporation of residual blocks and atrous spatial pyramid pooling (ASPP) module in the encoder part. The residual blocks help in extracting deep features from the data set while the ASPP module re-samples a given feature layer at multiple convolution sizes. This helps to extract the salient objects in multiple convolution sizes and extracts more prominent features from the given images. The proposed model SODU2-NET, has been evaluated by publicly available datasets such as CCD, DUTS, OMRON, DUTS-TE, HKU-IS, and Pascal. Furthermore, besides the publicly available data set the proposed scheme is tested on the real images collected at Central South University and its surroundings in Changsha, Hunan, China. The proposed model achieved better results in terms of mean absolute error results as compared to the state-of-the-art methods.

## Our contribution

1) A DL-based model SODU2-NET is proposed for automated saliency detection using residual blocks, and atrous spatial pyramid pooling (ASPP). The standard U-NET architecture is modified by adding the residual blocks inside the encoder architecture. Furthermore, ASPP is used inside each encoder which resulted in improved salient object detection. Adam optimizer is a stochastic gradient descent (SGD) variant used as the learning technique to train U-Net. Cross-entropy loss, a popular loss function for classification issues, is typically the one employed. Back propagation is used during the training phase to modify the network's weights and biases in order to minimize the loss function.

2) The proposed scheme SODU2-NET, is tested on the publicly available dataset (DUTSTE, DUTS- OMROM, DUTS-SOD, HKU-IS, and Pascal). Furthermore, besides these datasets, the proposed scheme is tested for real-time applications on a novel custom-collected image from real life. The existing schemes are not been able to perform well on the new data set, whereas the proposed scheme has achieved better performance on the custom collected data set.

3) We have also used multiple metrics to identify and compare the issue of SOD in old models. We performed thorough quantitative assessments to validate the efficacy of every component integrated into our novel model.

4) The learning algorithms use back propagation to calculate gradients and update model parameters during training. They play a important role in finding the optimal configuration of the U-Net model for accurate and efficient saliency detection.

The rest of the article is organized as follows; "Related Work" discusses the existing literature. In "Proposed Methodology" the proposed scheme is discussed whereas, in "Experimental Setup" the results and comparison of the proposed scheme SODU2-NET, with the existing schemes are discussed. Finally, "Experimental Results and Analysis" concludes the whole work.

## RELATED WORK

This section explains the existing work related to saliency detection. As compared to the other machine vision tasks saliency detection is a primary and the most challenging task due to its variable scale and an unknown number of categories within a particular input image. *Teng et al. (2024)* proposed LF Tracy model introduces a novel paradigm for enhancing salient object detection (SOD). Therefore, many DL techniques have been widely used for saliency detection in computer vision (*Ullah et al., 2020*). Deep neural networks, specifically convolutional neural networks (CNNs), are known for their excellent performance in this task by learning high-level representations from raw image data (*Ding et al., 2021*). Self-supervised co-salient object detection *via* feature correspondence at multiple scales introduced by *Chakraborty & Samaras (2024)*. These networks are trained on large datasets to predict the saliency map, which is a heatmap indicating the degree of importance of each pixel in the image. Some of the popular DL models for saliency object detection include DeepGaze (*Kümmerer, Theis & Bethge, 2014*), C-DSS (*Chen et al., 2020*), and DSS (*Zhang et al., 2020*). These models are designed to handle real-world images and videos with various objects, scales, and cluttered backgrounds also (*Dulam & Kambhamettu, 2023*).

In recent years, this problem is solved by *Pang et al. (2020)*, Using aggregate interaction modules. Self-attention components are embedded in each unit of the decoder part for efficient feature extraction. Consistency-enhanced loss is used to highlight the difference between the object and the background. The authors have used a benchmark dataset to evaluate the performance of the proposed model. *Zhuge et al. (2022)* have proposed a model that extracts and combines low and high-level features from the given data set. This model introduces three modules for feature extraction: DFA, ICE, and PWV. ICON model

was proposed and tested on seven benchmark datasets. *Mishra et al. (2020)* proposed pooling-based modules for saliency detection. A U-shaped architecture was built using a global guidance module (GGM) module on the bottom pathway. The Texture-Semantic Collaboration Network for ORSI Salient Object Detection proposed by *Li, Bai & Liu (2023)*. A feature aggregation module is used to pass important semantic features to the top-way path. MobileNetv2 (*Indraswari, Rokhana & Herulambang, 2022*) is used as an encoder that extracts features from the input dataset. The proposed model achieved higher results on mobile devices as well as edge detection problems also introduced a newer approach (*Song et al., 2023*).

*Wu, Su & Huang (2019)* have proposed a cascaded partial decoder (CPD) for accurate and fast salient object detection. The partial decoder is constructed in a framework that neglects the features with larger resolution within shallow layers for acceleration and by features integration of deep layers obtains a precise saliency map. Hence, direct utilization of saliency maps to iteratively optimize features in deep layers suppresses the distractors and enhances their representation ability. *Wang et al. (2019)* have proposed a salient object detection algorithm using the stacked attention network design for enhancing the representation ability of the network layers. Furthermore, this scheme has achieved better results in terms of detecting refined object boundaries that may get critical for applications that are related to geometric designs.

*Zeng et al. (2019)* has proposed a scheme that specifically works with low-resolution images. This novelty of the work was to use local high-resolution details with global semantic information to achieve better results for the high-resolution salience dataset. *Wu et al. (2019)* developed a mutual learning module, trained using a mutual learning approach, to improve the performance of the proposed algorithm. *Zhao et al. (2019)* have discussed another scheme that mainly focuses on the edge information of the detected objects. They presented a network for exploring edge information (EGNet) for the detection of salient objects.

## Alterations in U-net

The U-net architecture is a specialized convolutional neural network design tailored for the task of image segmentation. Its architecture is made up of an encoder and a decoder that are linked by a bottleneck layer. The encoder is responsible for acquiring high-level features from an input image. whereas the decoder learns to reconstruct the segmentation map output. The purpose of saliency detection is to identify and segment the most visually distinct objects in a real image. The distinctive objects, also known as salient objects, are those that stand out from their surroundings due to their color, texture, shape, or other visual characteristics.

Because it can effectively capture the hierarchical and multi-scale features of an image, UNET is well-suited for salient object detection. Furthermore, the architecture's use of skip connections allows for the combining of both high-and low-level features, which increases the salient object's accuracy and improves the accuracy of salient object detection even further. Because it can effectively capture the hierarchical and multi-scale features of an image, UNET is well-suited for salient object detection. Furthermore, by employing skip

connections within the architecture, the integration of both low-level and high-level features becomes possible. This integration significantly enhances the precision of salient object detection.

*Kumar et al. (2022)* introduced a salient object recognition technique rooted in DL networks, which upholds image data accuracy within the mid and low regions. This method creates a coarse saliency map for the full target image using a DL model. The map is then adjusted using low to mid-level details specific to the image. The author also adopts a U-Net architecture to detect prominent items. Pixel-by-pixel prediction of the saliency map minimizes the loss of low-level visual information. The findings demonstrate that their system consistently outperforms competing methods for identifying salient objects, leading to higher precision and recall rates.

In this study, *Han, Li & Dong (2019)* endeavors to predict the saliency map of the image by incorporating an edge convolution constraint into an adapted U-Net architecture. It can limit information loss by fusing the features of several layers in the proposed network structure. As opposed to the CNN- based models, our SalNet predicts the saliency map at the pixel level. Additionally, a novel loss function is introduced in this research, utilizing image convolution to introduce an L1 constraint on the edge details of both the saliency map and the ground truth. This innovation enhances the network's ability to capture crucial edge information pertaining to object delineation. The conclusive experimental results validate the efficacy of our SalNet model in the salient object identification task and is competitive when measured against 11 leading models.

*Chen et al. (2023)* have discussed an Adaptive Fusion Network (AFNet) to solve the problems related to the noise within the depth of the colored images, in context to saliency detection. The authors have proposed a triplet encoder network compromising three sub-networks. These networks are responsible for processing. The study involves three distinct sub-networks designed to handle RGB, depth, and fused features, respectively. These sub-networks are then interconnected to establish a grid net, facilitating the collaborative enhancement of these multi-modality features. Furthermore, an inventive Multi-modality Feature Interaction (MFI) module is introduced to harness the synergistic information between depth and RGB modalities. This adaptive fusion enhances the integration of multi-modality features. Additionally, the study presents the Cascaded Feature Interwoven Decoder (CFID), which adeptly capitalizes on harmonized information from multi-level features and sequentially refines their processing, ultimately yielding improved outcomes. Although these algorithms have performed very well in terms of accuracy as well as the mean absolute error. However, when these algorithms are tested for the new dataset which contains images from Changsha city in Hunan province of China, the performance in terms of F1-score, S measure, E measure, and mean absolute error were low. After that, we observed the issue and proposed changes to improve the accuracy of the existing algorithm. Therefore, in this work, the standard architecture of the U-net model is modified for salient object detection.

# PROPOSED METHODOLOGY

In this work, we have used the standard U-Net architecture along with residual blocks that were proposed in resnet architecture and the ASPP technique for salient object detection. In the coming subsection, we are discussing these blocks one by one, and finally, in the last subsection the architecture of the proposed model is discussed.

## Proposed model for saliency object detection

In this study, a DL approach is used for the detection of salient objects in images. The proposed model is a combination of U-Net (*Zhang et al., 2023*), ResNet (*Xu et al., 2022*), and DeepLab ASPP (*Movahedi & Elder, 2010*). First, the full feature fusion module is used that help to extract features from multiple scales. After that, Squeeze and Excitation (SEM) is used to improve spatial feature maps. After resizing all the features, an ASPP modules is used to capture features from multi-scale kernerl sizes. The standard architecture of U-Net is used which consists of encoder decoder-based architecture. The encoder blocks are used for feature extraction while the decoder part is used for final salient object detection. The U-Net encoder component is modified by adding residual blocks and ASPP. Residual blocks are used to extract the deep features from the images. Deep feature extraction helps to preserve the important features that can be lost due to convolution operations. ASPP module extracts feature using different convolution sizes. Different convolution size helps to extract important information from multiple image sizes. The architecture of the proposed model is shown in Fig. 1.

1) Encoder-decoder architecture: U-Net adopts an encoder-decoder architecture. The network comprises a contracting path called the encoder and an expanding path known as the decoder. The encoder captures essential information and extracts features from the input image, whereas the decoder aids in generating a high-resolution segmentation map.

2) Skip connections: U-Net incorporates skip connections between the contracting and expansive paths. These connections allow information from higher-resolution feature maps in the contracting path to be directly passed to corresponding stages in the expansive path. Skip connections help in preserving detailed spatial information and facilitate better segmentation results.

3) Fully convolutional: U-Net is a fully convolutional network, which means it operates on the entire image rather than fixed-size patches. This property makes it suitable for handling images of different sizes and enables efficient training and inference.

4) Effective for small datasets: U-Net performs well even when training data is limited. This is achieved through data augmentation techniques such as random flips, rotations, and elastic deformations, which artificially increase the size of the training set.

5) Highly successful in biomedical imaging: U-Net has found extensive use in biomedical imaging applications, such as segmenting organs, tumors, and cell structures. Its ability to capture fine details and handle complex shapes makes it well-suited for such tasks.

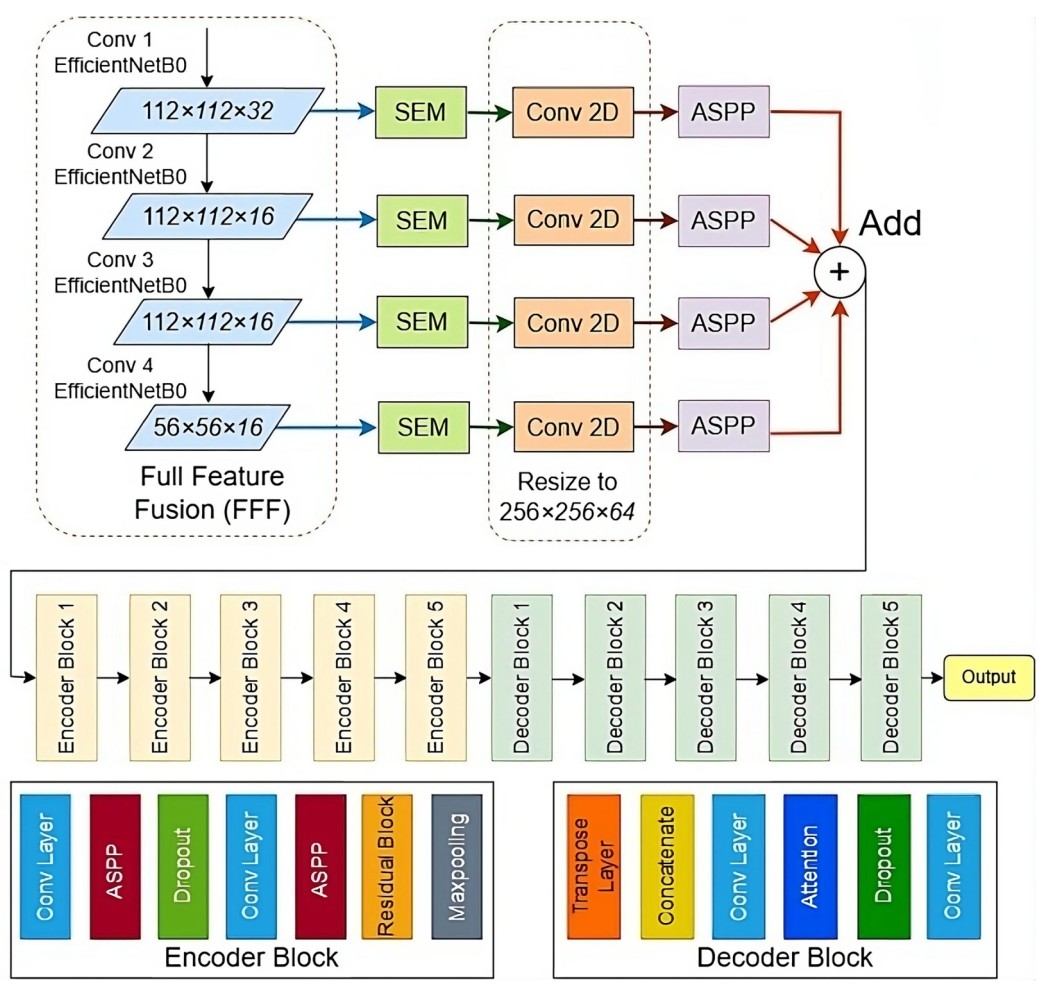

**Figure 1** The proposed SODU2-NET model for salient object detection.

6) Modifications and variations: The U-Net architecture has been widely adopted and serves as a foundation for further research and improvements. Numerous variations and extensions of U-Net have been proposed to address specific segmentation challenges, such as nested U-Nets, attention mechanisms, and multi scale approaches.

The expanding path is also similar to a traditional CNN, In this process, a sequence of convolutional and up-sampling layers are employed to decrease the number of channels, concurrently enhancing the spatial resolution of the output and diminishing the number of channels. This path is called expanding because the spatial resolution increases while the number of channels decreases.

## Full feature fusion (FFF)

In machine learning and neural networks, feature fusion refers to the process of combining features extracted from different sources or layers of a model to create a more informative representation of the input data (*Rashid et al., 2020*). Efficient NetB0 is a type of neural

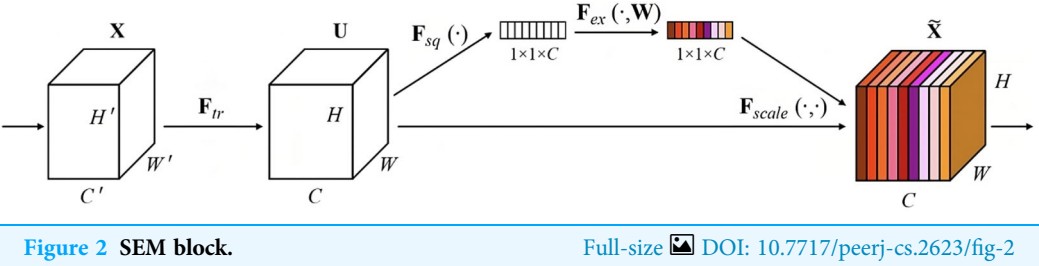

**Figure 2** SEM block.     

network architecture designed for image classification tasks. It is known for its efficiency and high performance in terms of accuracy and computational resources. In this study, Efficient NetB0 is used for FFF for extracting multi-scale features from the input dataset the image SEM block is shown in Fig. 2.

## Squeeze and Excitation block (SEM)

Squeeze-and-Excitation (SE) blocks are a type of neural network module that can be plugged into any convolutional neural network (CNN) to improve its performance. They do this by explicitly modeling channel relationships and channel inter-dependencies in CNNs.

SE blocks work by first squeezing each channel of a feature map into a single numeric value using global average pooling. This is followed by an excitation operation, which uses two fully connected layers to generate a set of weights for each channel. These weights are then used to re-scale the feature maps, giving more importance to the channels that are most relevant to the task at hand.

SE blocks have been shown to be effective at improving the performance of CNNs on a variety of tasks, including image classification, object detection, and semantic segmentation. They are also very efficient, adding minimal computational overhead to CNNs.

## Standard architecture of U-Net model

The U-Net architecture (*Zhang et al., 2023*) is a DL framework devised for image segmentation pur-poses, encompassing tasks like medical image analysis, semantic segmentation, and object detection. Developed by Olaf Ronneberger, Philipp Fischer, and Thomas Brox in 2015, U-Net comprises two primary components: a contracting and an expanding path, depicted in Fig. 1. Drawing inspiration from VGGNet, the contracting segment employs stride convolutions, max pooling, and ReLU activation functions to diminish spatial dimensions while augmenting the feature map count. The expanding path uses transposed convolutions, concatenation with the feature maps from the contracting path, and ReLU activation functions to restore the spatial dimensions and refine the features.

The contracting path is similar to a traditional CNN, where a series of combinations of convolutional and max pooling layers is used to increase the number of channels while decreasing the spatial resolution of the input image. This path is called contracting because the spatial resolution decreases while the number of channels increases.

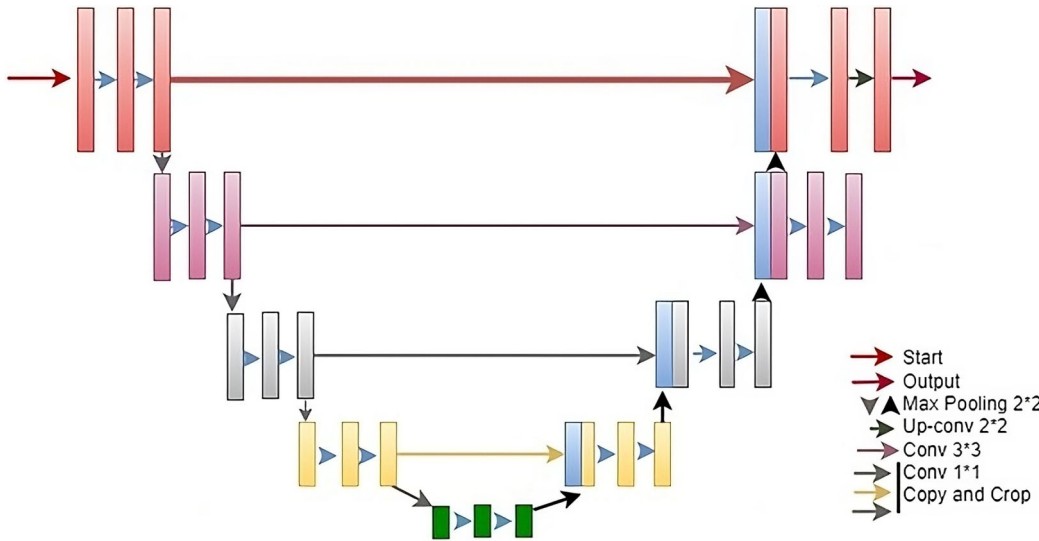

**Figure 3  The standard architecture diagram of U-Net.**

The expanding path is also similar to a traditional CNN, where a series of convolutional and up-sampling layers are applied to lower the number of channels while increasing the spatial resolution of the output while decreasing the number of channels. This path is called expanding because the spatial resolution increases while the number of channels decreases. As can be seen from the Fig. 3, in the U-Net encoder stage, a 3 × 3 convolutional kernel is used for convolution followed by a ReLU activation function and a pooling with a scale of 2 × 2 for down-sampling. This operation allows the feature channels to be expanded to twice the size of the previous layer. In the decoder stage, a 2 × 2 convolution kernel is used for up-sampling, where each up-sampled channel is half the size of the previous one, which is then spliced with the feature maps (after cropping) from each stage of the encoder stage to obtain a channel of the same size as the encoder stage. In the last step of the network, a 1 × 1 convolutional layer is added for classification. As U-Net applies more information in the decoder stage by fusing the feature maps from the encoder stage, it can effectively preserve the edge information of the image. The U-Net architecture was first applied to the detection of cell-based images and has since been widely used in medical image segmentation. The U-Net model is not only suitable for small datasets but also has good performance in image segmentation. Therefore, this article chooses to use the U-Net model as the base network to build a new model, and applies the U-Net model to the field of image saliency detection, and builds a better image saliency detection model with better performance.

## Deep feature extraction using residual blocks

Residual blocks are building blocks used in residual neural networks (ResNets) as shown in Fig. 4, a type of deep con-volutional neural network. Residual blocks serve the purpose of alleviating the challenge of vanishing gradients encountered in deep neural networks. This predicament arises when error gradients diminish significantly as they traverse through

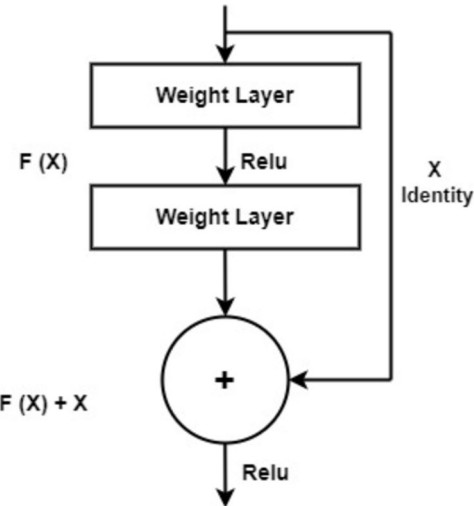

**Figure 4 The standard architecture diagram of residual blocks.**

numerous layers, impeding effective learning within the network. A residual block is structured with two or more layers, often convolutional layers, interconnected by shortcut connections. These shortcuts enable the bypassing of one or more layers, as shown in Fig. 2. The presence of shortcut connections characterizes a residual block, which comprises multiple layers, often convolutional layers, along with shortcut connections that bypass one or more of these layers, as illustrated in Fig. 2. These shortcut connections facilitate the residual block in learning the residual mapping connecting the input and output, rather than the actual mapping. The residual mapping is the discrepancy between the input and output, which is simpler to learn and addresses the vanishing gradient issue. Residual blocks (*Bello et al., 2021*) are used in deep ResNets, where authors are stacked together to form very deep neural networks. As a result, the network can acquire intricate representations, leading to exceptional performance on diverse computer vision tasks, including image classification and object detection.

## Atrous spatial pyramid pooling

ASPP (*Chen et al., 2017*) is a technique used in deep convolutional neural networks (CNNs) for the semantic segmentation of images. ASPP is used to incorporate context information into the feature maps generated by CNNs and improve the segmentation results. ASPP consists of multiple parallel branches, each with a different dilation rate, which is used to capture context information at different scales as shown in Fig. 5. The branches use dilated convolutions, which have a larger receptive field compared to normal convolutions, to feature extracted from the feature maps. The features from all branches are concatenated and fed into a fully connected layer to produce the final segmentation map. By using multiple parallel branches with different dilation rates, ASPP is able to capture context information at different scales, which is crucial for image segmentation tasks where objects can have varying sizes and shapes. ASPP has been shown to be effective

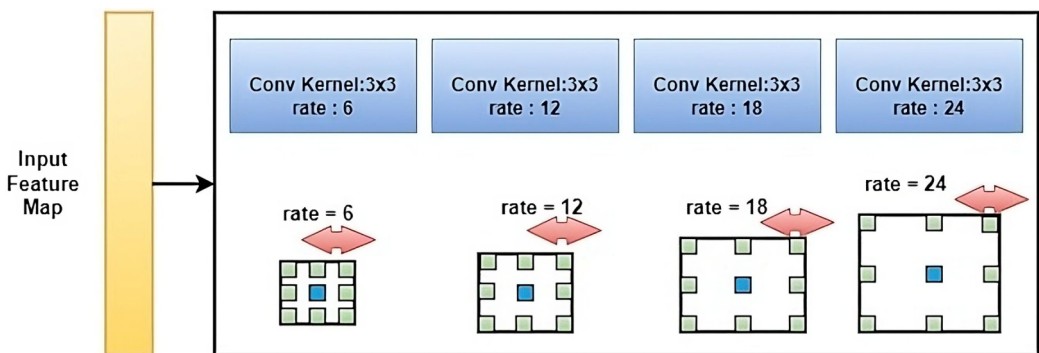

**Figure 5 The standard architecture diagram of ASPP.**

in improving the performance of CNNs on various semantic image segmentation benchmarks.

## EXPERIMENTAL SETUP

In this section, the experimental setup, results, and comparison of the proposed SODU2-NET are discussed.

### Dataset definition
#### DUTS dataset

Dataset for Understanding and Testing Saliency (DUTS) is a large-scale saliency dataset for evaluating computer vision algorithms for visual saliency prediction (*Wang et al., 2017*). The dataset comprises 10,000 high-resolution images featuring a diverse array of objects, scenes, and image types. It is extensively used as a benchmark dataset for evaluating saliency prediction algorithms in the fields of computer vision and human perception. The data set is one of the widely used data sets for saliency detection as discussed in *Lee, Shin & Han (2022)*, *Zhang et al. (2023)*, and *Xu et al. (2022)*. The sample images from the DUTS dataset are shown in Fig. 6.

#### Salient objects dataset

The salient objects (SO) dataset is a large-scale image dataset for evaluating saliency detection algorithms in computer vision (*Movahedi & Elder, 2010*). The dataset contains a total of 10,000 images, and each image has been manually annotated to mark the most visually distinctive and attractive regions. The annotations are used as ground truth to evaluate the performance of saliency detection algorithms. The SO dataset is a commonly used benchmark in computer vision research and has been used in several studies (*Song et al., 2022*; *Fan et al., 2022*; *Huang et al., 2022*) to compare and evaluate different saliency detection algorithms, some of the images from the SO dataset are shown in Fig. 7.

#### DUT OMRON dataset

The DUT OMRON (*Jiang et al., 2011*) Image Dataset is a large-scale image dataset for evaluating visual saliency prediction algorithms in computer vision. The dataset was created by the Omron Corporate Research and Development Centre and Dongguk University in South Korea. It contains 5,168 high-resolution images, covering a wide range

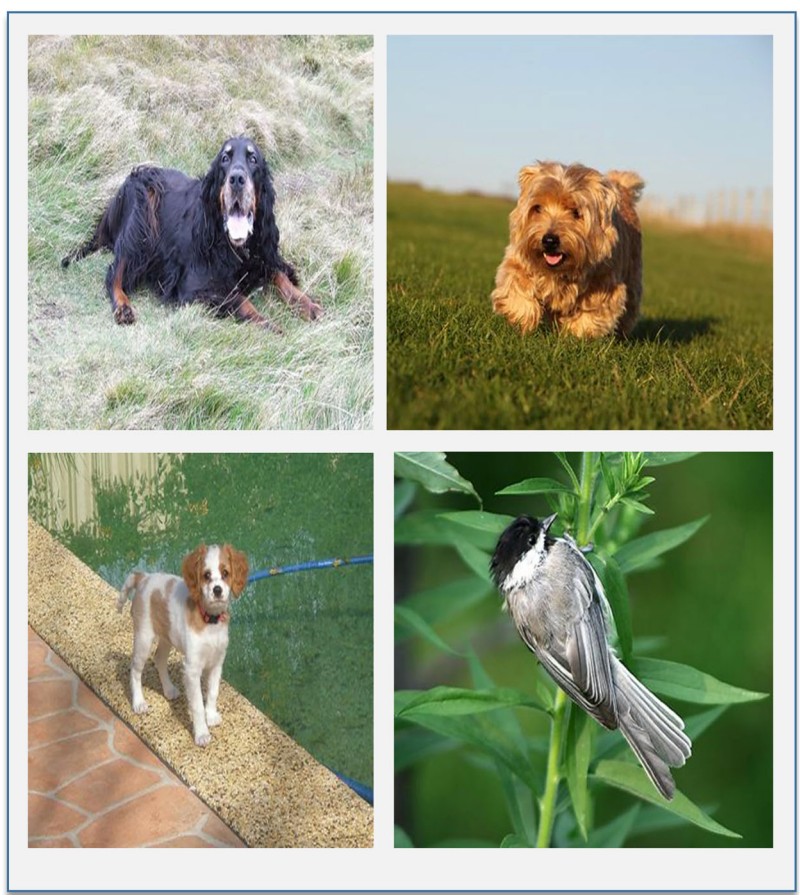

**Figure 6 The sample images of the DUTS dataset.**

of objects, scenes, and image types. The images in the dataset have been manually annotated to mark the most visually distinctive and attractive regions, which serve as the ground truth for evaluating the performance of saliency prediction algorithms. The DUT-OMRON dataset is widely used in computer vision research as a benchmark for evaluating and comparing different saliency prediction algorithms as discussed in *Lee, Shin & Han (2022)*, *Zhang et al. (2023)*. Some of the images from the DUT OMRON dataset are shown in Fig. 8.

### Hong Kong University of Science and Technology Information Systems (HUK-IS) dataset

The Hong Kong University of Science and Technology Information Systems (HUK-IS) dataset is a large-scale image dataset for evaluating visual saliency prediction algorithms in computer vision (*Li & Yu, 2015*). The dataset was created by researchers at the Hong Kong University of Science and Technology and contains 4,447 high-resolution images, covering a wide range of objects, scenes, and image types. The images in the dataset have been manually annotated to mark the most visually distinctive and attractive regions, which serve as the ground truth for evaluating the performance of saliency prediction algorithms. The HKU-IS dataset is widely used in computer vision research as a benchmark for

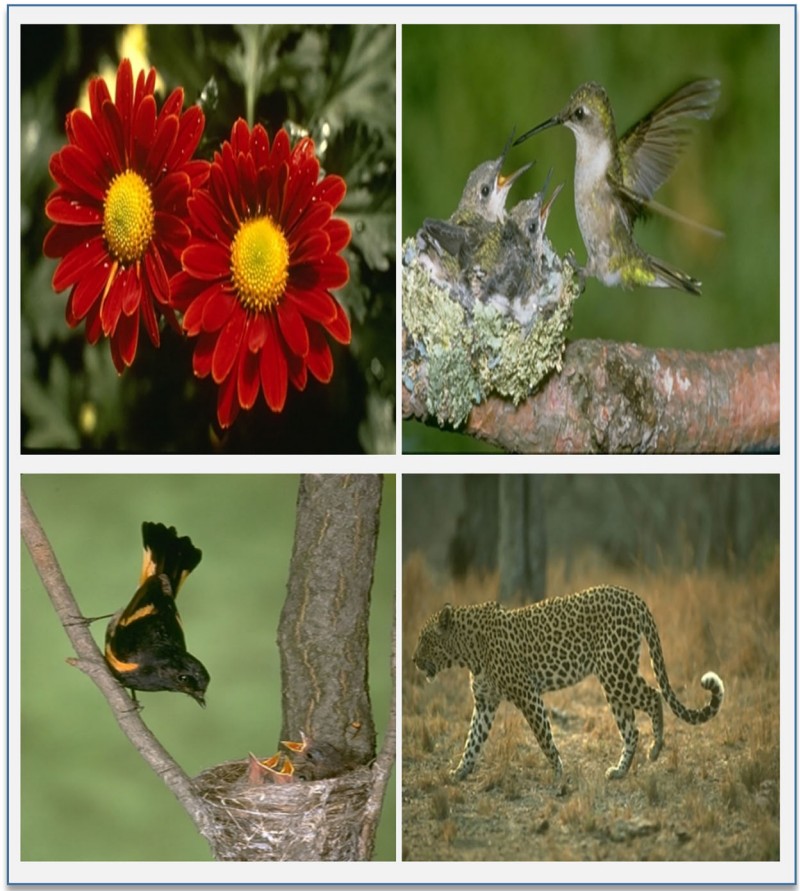

**Figure 7 The sample images of the SOD dataset.**

evaluating and comparing different saliency prediction algorithms as discussed in *Bao, Dai & Elsaddik (2022)*, *Xu et al. (2022)*, and *Liu et al. (2022b)*. The sample images from the HKU dataset are shown in Fig. 9.

### PASCAL saliency (PASCAL-S) dataset

The PASCAL saliency (PASCAL-S) is a dataset for evaluating saliency detection algorithms in computer vision (*Dong et al., 2021*). It is a subset of the PASCAL Visual Object Classes (VOC) dataset, which contains a total of 850 high-resolution images. Each image in the PASCAL-S dataset has been manually annotated to mark the most visually distinctive and attractive regions, which serve as the ground truth for evaluating the performance of saliency detection algorithms. The PASCAL-S dataset is widely used in computer vision research as a benchmark for evaluating and comparing different saliency detection algorithms as discussed in *Liu et al. (2022a)*, *Xu et al. (2022)*, *Yang, Chen & Deng (2022)*. Figure 10 shows the sample images from PASCAL dataset.

### Salient objects of Changsha China

The proposed model SODU2-NET, is evaluated on salient object of Changsha, Hunan, China dataset. The dataset is captured using an iPhone 11 with a 12-megapixel camera.

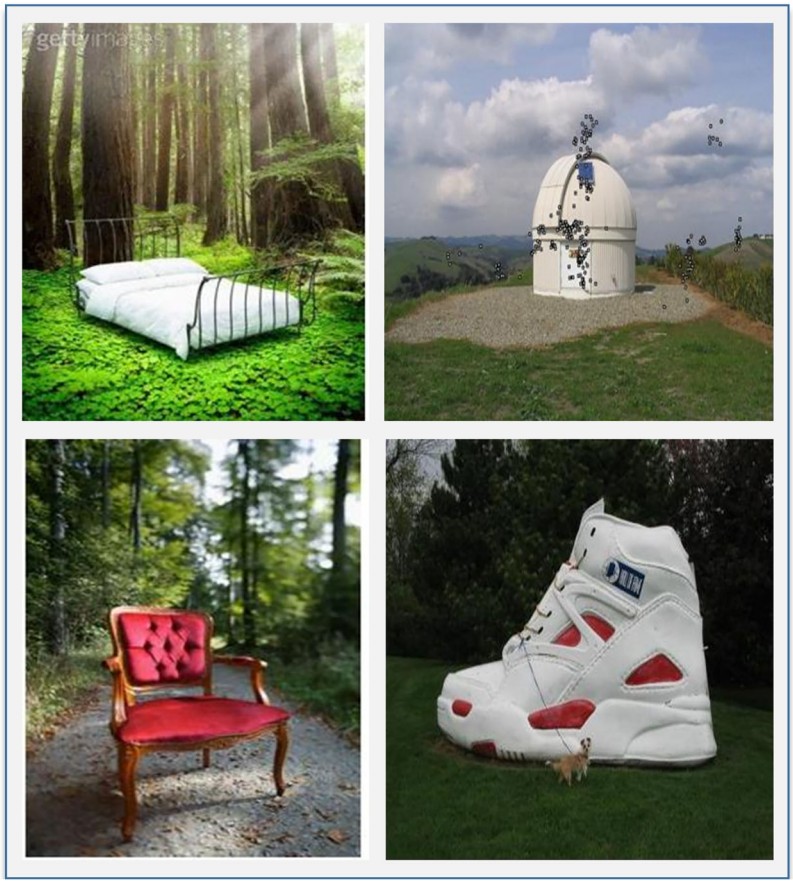

**Figure 8  The sample images of the DUT-OMRON dataset.**

Multiple salient objects are captured and a dataset is created to use in this study. The sample images of the developed dataset are shown in Fig. 11. It is evident from the that indoor as well as outdoor images of different objects are collected in different light resolutions so that the generality of the proposed scheme can be tested. The overall dataset contains 622 images that were collected from the campus of the central south university and across the city.

## Model training parameters

Throughout the machine learning model's training phase, several parameters are specified to govern the learning trajectory and enhance overall performance. These encompass the learning rate, batch size, epochs, optimizer, loss function, and early stopping strategy. The dataset is bifurcated into two segments, with 80% allocated for training and the remaining 20% reserved for testing the proposed model. Uniform resizing to dimensions of $256 \times 256$ is applied to all images, subsequently fed into the network to facilitate model training. The training stage encompasses assorted data augmentation techniques, including random horizontal flipping, random cropping, and scaling, aimed at mitigating the risk of overfitting.

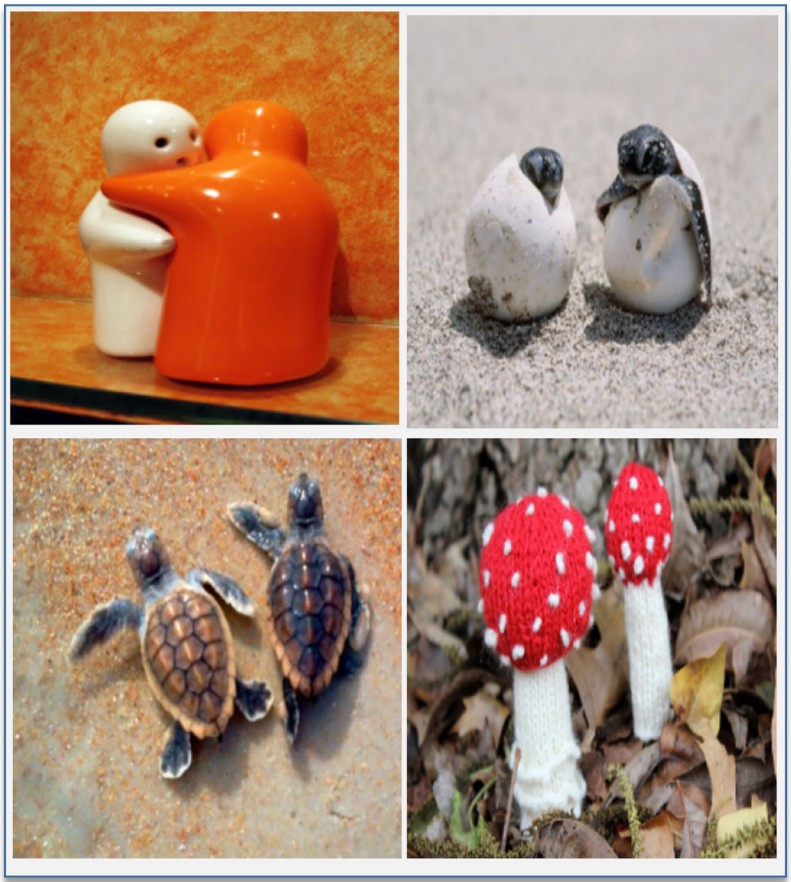

**Figure 9** The sample images of the HKU dataset.

The Kaggle notebook serves as the training environment for the SODU2-NET model, leveraging a system equipped with 16 GB RAM and an NVIDIA GTX 2080 GPU. The model is initiated with a base learning rate of 0.0001, while a batch size of 8 and 100 epochs contribute to the training process. Utilizing the Adam optimizer (*Zhang, 2018*), the training incorporates early stopping mechanisms: a learning rate reduction of 0.1 if no accuracy improvement is observed within the first 10 epochs, and termination if accuracy stagnates for 15 consecutive epochs. Cross-entropy is embraced as the chosen loss function.

Notably, the entire training process spans a duration of 1 h and 8 min, with each individual prediction requiring 1.2 s for completion.

### Ablation study

To assess the effectiveness of our proposed method, we conducted detailed ablation experiments, systematically evaluating the contribution of each component. We present the results of two sets of experiments: structural ablation experiments and loss function ablation experiments.

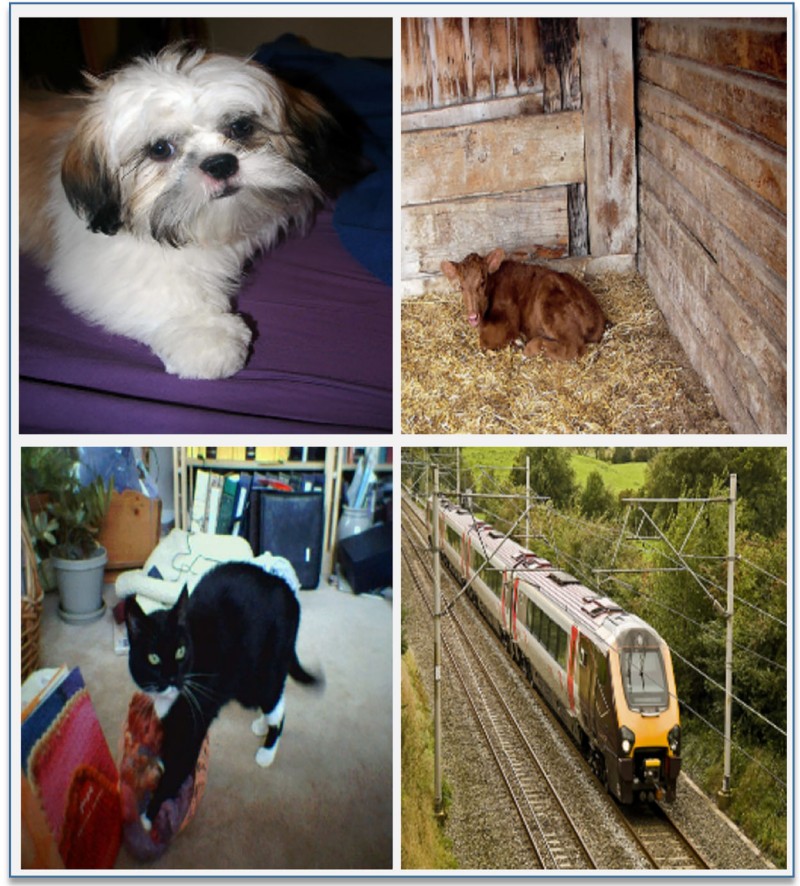

**Figure 10 The sample images of the PASCAL dataset.**

1. Structural ablation experiments:

In these experiments, we quantitatively compared the results obtained using different modules. We added separate instances of the multi-level feature aggregation module, multi-scale information extraction module, and boundary extraction module to the Baseline. Subsequently, we examined combinations of these modules for further experimentation.

The results in Table 1 demonstrate significant performance improvements across all modules. Notably, the multi-scale information extraction module exhibits the most promising results, achieving a considerable improvement in various metrics compared to the baseline. Combining multiple modules leads to further enhancements, with the best performance observed when all three modules are utilized together.

2. Loss function ablation experiments:

In these experiments, with all modules equipped, we evaluated the effectiveness of single loss functions and combinations of two loss functions. The results in Table 2 show that the outer structure loss demonstrates a substantial improvement, particularly when combined with another loss function. This combination achieves the most favorable results across different evaluation metrics.

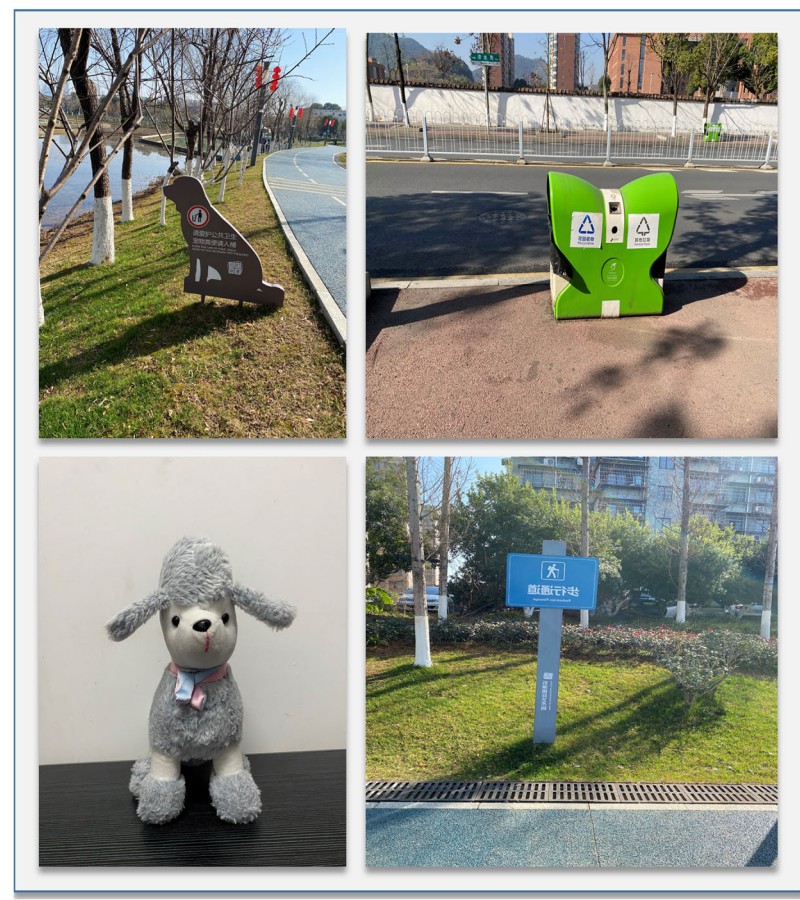

**Figure 11 Sample images of dataset capture in Changsha, Hunan, China.**

Table 1 Performance improvement of different modules.

| Module | Metric improvement |
| --- | --- |
| Baseline | – |
| Multi-level feature aggregation | Moderate |
| Multi-scale information extraction | Significant |
| Boundary extraction | Moderate |
| Multi-level + Multi-scale | High |
| Multi-level + Boundary | Moderate |
| Multi-scale + Boundary | High |
| Multi-level + Multi-scale + Boundary | Highest |

Table 2 Effectiveness of loss functions and combinations.

| Loss function combination | Performance improvement |
| --- | --- |
| Single loss (Outer structure) | Substantial |
| Single loss (Other) | Moderate |
| Outer structure + Other | Most favorable |

These ablation experiments highlight the importance of each component in our proposed method. The structural ablation experiments underscore the significant performance gains achieved by incorporating the multi-scale information extraction module, while the loss function ablation experiments emphasize the effectiveness of combining multiple loss functions for improved salient object detection accuracy.

## EXPERIMENTAL RESULTS AND ANALYSIS

In this section, the results of the proposed SODU2-NET, are discussed for different datasets. Table 3 shows the results of the proposed scheme on different datasets. The proposed scheme has achieved better results in terms of accuracy, precision, recall and F1-score.

- Goal (G): To evaluate the performance of the proposed SODU2-NET model for salient object detection.
- Question (Q1): What are the results of the proposed SODU2-NET model on different datasets?
- Question (Q2): What are the evaluation metrics commonly used to assess the SODU2-NET, model's performance for salient object detection?
- Question (Q3): What is the mean absolute error (MAE) curve?

### Results (R1)

The proposed SODU2-NET model achieved better results in terms of accuracy, precision, and F1-score as shown in Figs. 14–16. The SODU2-Net model, designed specifically for salient object detection, accurately identified and segmented salient objects in images.

### Results (R2)

Accuracy focuses on the overall correctness of the predicted saliency map, Precision measures the accuracy of the salient object segmentation, recall measures the ability to correctly identify salient objects, and F1-score combines precision and recall for an overall evaluation of model performance.

### Results (R3)

MAE was used to calculate the average pixel-wise absolute difference between the predicted saliency map and the ground truth. Figure 12 shows the MAE.

### Evaluation metrics

The U2-Net model is a DL architecture designed for salient object detection, which aims to identify and segment salient objects in images. When evaluating the performance of the U2-Net model, several evaluation metrics commonly used in computer vision tasks is employed for evaluating model performance.

Accuracy: Accuracy focuses on the overall correctness of the predicted saliency map rather than the localization and segmentation of salient objects.

**Table 3 Efficiency comparison of different models with the SODU2-NET model.**

| Models | Precision | Recall | F1 | Accuracy | References |
|--------|-----------|--------|-----|----------|------------|
| SODU2-NET | 98% | 97% | 97% | 96% | Proposed |
| FCN | 59% | 56% | 57% | 78% | *Girisha et al. (2019)* |
| SqueezeNet | 89% | 89% | 89% | 75% | *br Tarigan, Gunawan & Hayadi (2023)* |
| Deep Lab | 80% | 93% | 86% | 82% | *Chen et al. (2021)* |
| Mask R-CNN | 60% | 69% | 69% | 91% | *Suh et al. (2021)* |

| Input Image | Background Remove | Bounding Box | Saliency Map |
|-------------|-------------------|--------------|--------------|

**Figure 12 Saliency map.**

Precision: Precision quantifies the ratio of accurately predicted salient pixels (true positives) in relation to the entirety of pixels forecasted as salient (true positives + false positives). This metric offers insight into the precision of salient object segmentation.

F1-score: The harmonic mean of recall and precision is the F1 score. It integrates both metrics to give a comprehensive assessment of the model's performance, taking into account both precision and recall simultaneously.

MAE: MAE calculates the average pixel-wise absolute difference between the predicted saliency map and the ground truth saliency map. It provides an overall measure of the dissimilarity between the predicted and ground truth salient regions result of our model MAE is shown in Fig. 17.

Confusion matrices: Confusion matrices are typically used in classification tasks to assess the models' performance, an analysis is conducted by comparing the ground truth labels and the anticipated labels. The confusion matrix's cells each represent the quantity of instances that were classified into a particular class.

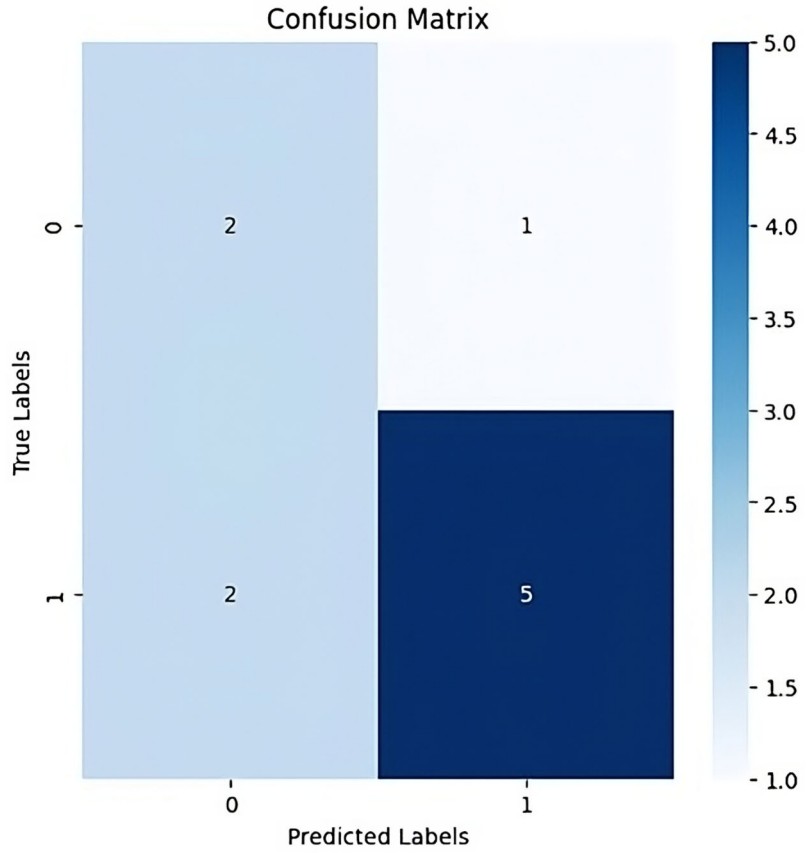

**Figure 13  Confusion matrix.**               

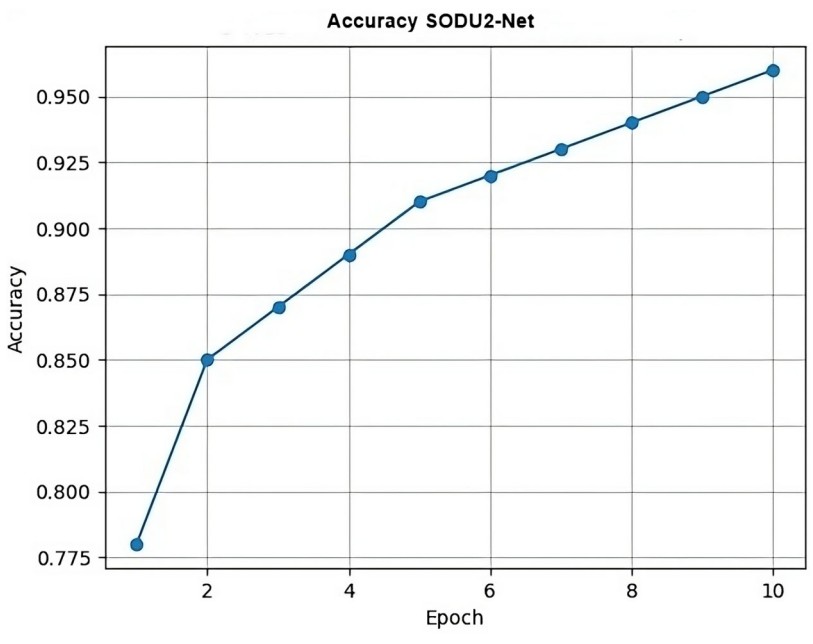

**Figure 14  Accuracy.**               

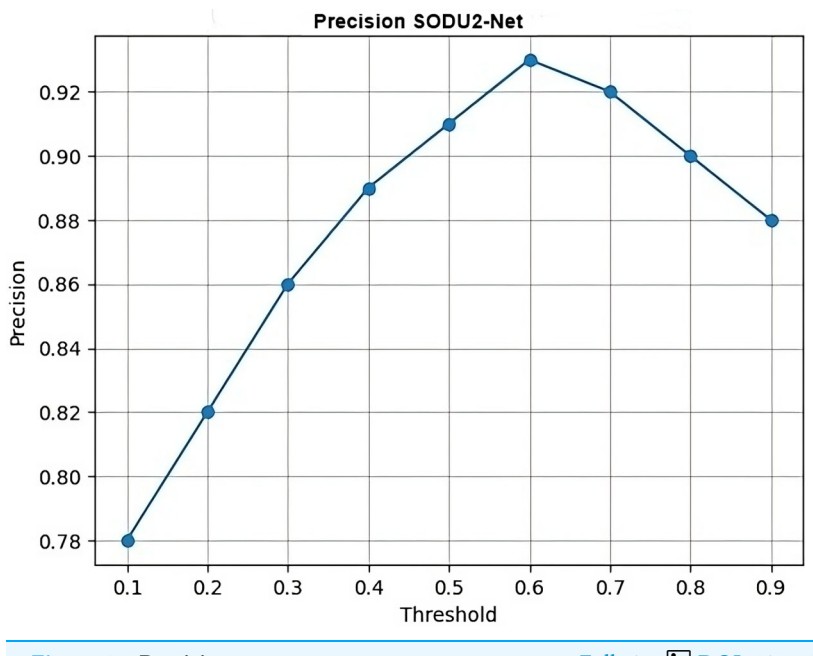

**Figure 15 Precision.**   

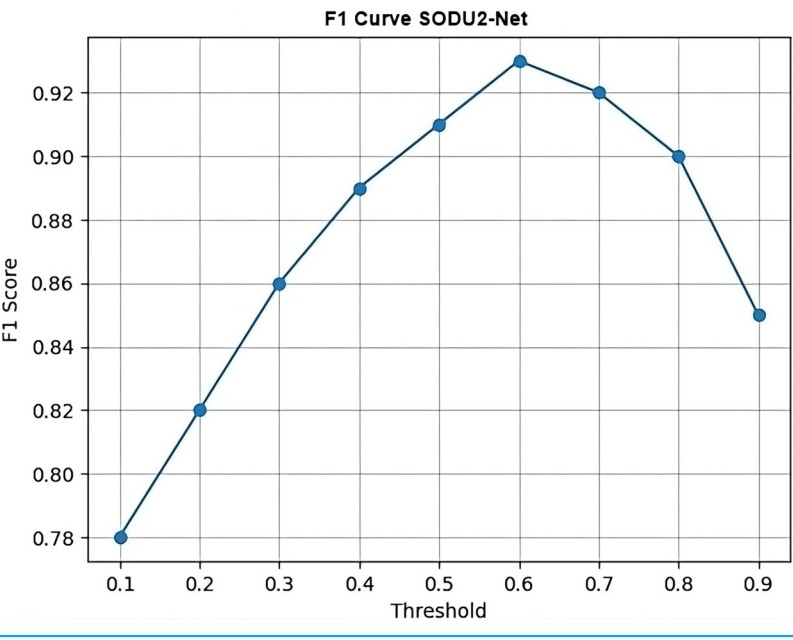

**Figure 16 F1 curve.**   

Figure 13 shows the process of salient map by background removal and bounding box creation which leads to salient feature highlighting.

## About evaluation index

The proposed SODU2-NET, model is typically trained using a variation of the standard back propagation algorithm with gradient descent. The description of the learning algorithm shown in 12 for training proposed model are as under:

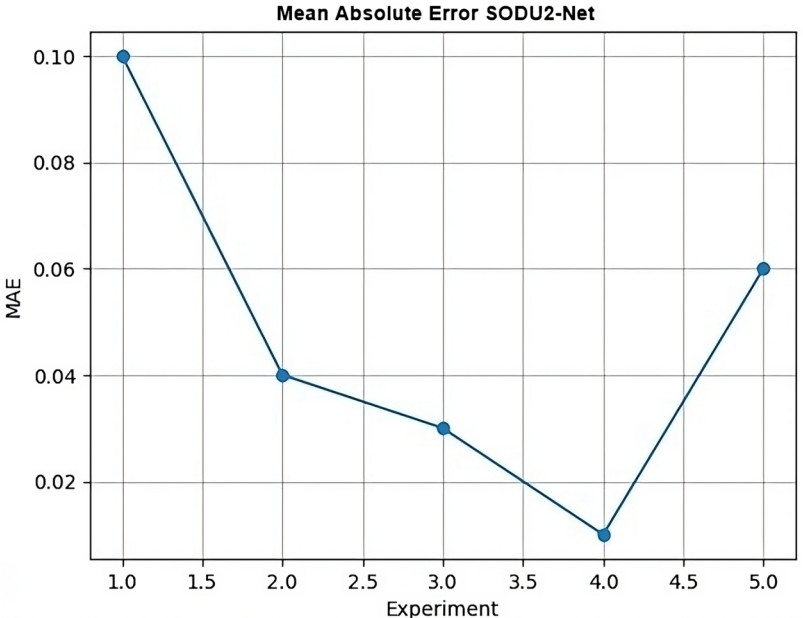

**Figure 17 Mean absolute error.**     

1) Data preparation: Training dataset were prepared, which consists of input images and corresponding ground truth segmentation masks. Ensure that the images and masks are appropriately Pre-processed and normalized for training.

2) Network initialization: Initialize the U-Net model with random weights. The weights can be initialized using techniques like Xavier or He initialization to promote stable and efficient training.

3) Forward propagation: Pass the input images through the proposed SODU2-NET, model in the forward direction. The encoder path processes the input images, extracting hierarchical features, while the decoder path gradually up samples the feature maps to generate the segmentation predictions.

4) Loss calculation: Evaluate the anticipated segmentation masks in contrast to the reference ground truth masks by employing a suitable loss function. Common choices for image segmentation encompass loss functions like pixel-wise cross-entropy loss or adaptations of the Dice loss. This loss function serves to quantify the disparity between the projected and actual masks, offering an assessment of the model's efficacy.

5) Back propagation: Determine the loss function's gradients with respect to the model's parameters. This involves propagating the errors from the loss function backward through the network and adjusting the model's weights *via* gradient descent. The gradients can be computed effectively through methods like automatic differentiation.

6) Parameter update: Revise the model's parameters (weights) using an optimization technique like stochastic gradient descent (SGD) or its derivatives (*e.g.*, Adam, RMSprop). This adjustment process governs the weights towards minimizing the loss function, facilitating the model's learning from the provided training data.

7) Iteration: For more iterations or epochs, repeat steps 3 to 6, where each iteration entails forward propagation, loss calculation, back propagation, and parameter update. During training, the model gradually learns to improve its segmentation performance by minimizing the loss function.

8) Validation and evaluation: Periodically evaluate the model's performance on a separate validation dataset to monitor its generalization ability and prevent overfitting. Use appropriate evaluation metrics, such as Intersection over Union (IoU) or pixel accuracy, that assess model segmentation accuracy.

9) Testing: After the model is trained and validated, it becomes applicable for inference on novel, unseen images. Apply the test images to the trained SODU2-NET model, thereby obtaining segmentation predictions.

To study the existing techniques and find out the research gap. There is a lot of work done in salient object detection but, As a result of the models' inability to extract the most pertinent data, there is still room for development. and visually striking parts of an image or video and detect small objects accurately. The proposed model is modified standard architecture of U-Net by adding the residual blocks and ASPP module in the encoder part. The residual blocks help in extracting deep features from the data set while the ASPP module re-samples a given feature layer at multiple convolution sizes. This helps to extract the salient objects in multiple convolution sizes and extracts more prominent features from the given images. The accuracy of model is 0.94 in comparison with FCN 0.78, Deep Lab 0.82, Mask R-CNN 0.91, and Squeeze Net 0.75.

## Accuracy

In Fig. 14, epoch represents the x-axis values (epochs) and accuracy represents the y-axis values (accuracy scores) for each epoch which generate a simple line plot with markers representing the accuracy values at each epoch.

## Precision

In Fig. 15, thresholds represent the x-axis values (thresholds) and precision represents the y-axis values (precision scores) for each threshold which generates a simple line plot with markers representing the precision values at each threshold.

## F1 curve

Figure 16 calculates the F1 scores using the precision and recall values and then generates a simple line plot with markers representing the F1 scores at each threshold. The x-axis represents the thresholds, and the y-axis represents the F1 scores. F1 curve is plotted by varying the threshold for classification and computing the corresponding F1 score at each threshold. It shows the SODU2-NET performance changes as the threshold for classifying a sample as positive or negative is adjusted. F1-score is calculated as in formula shown below.

$$f1\_scores = \frac{2(p.r)}{p+r} \quad \text{for} \quad p, r \quad \text{in} \quad \text{zip precision, recall} \tag{1}$$

## MAE

MAE is typically a single value that represents the average absolute difference between the true labels and predicted labels. Therefore, it's not common to generate a graph specifically for the MAE. However, in Fig. 17 it can visualize the changes in MAE over different experiments. MAE is a measure of the average absolute difference between the true values and the predicted values. It is calculated using the following formula:

$$MAE = \frac{1}{n} \sum_{i=1}^{n} |y_i - \hat{y}_i|$$

where:

MAE is the mean absolute error.

n is the number of samples.

y represents the true values.

$\hat{y}$ represents the predicted values.

—.— denotes the absolute value.

To calculate the MAE, you take the absolute difference between each true value and its corresponding predicted value, sum them up, and then divide by the number of samples.

## Comparative analysis

Comparative analysis of U-Net model with other DL models in salient object detection reveals its strength in preserving fine details and capturing object boundaries. FCN struggles with object boundaries, DeepLab excels in capturing global context but may lose fine details, mask region-based convolutional neural network (mask R-CNN) focuses on instance segmentation, and SqueezeNet offers efficiency at the cost of some accuracy. The choice of model depends on the specific requirements of the application, balancing accuracy, computational resources, and the desired level of detail in salient object detection Comparison s shown in Fig. 18.

Comparative analysis of the U-Net model with other DL models in performing salient object detection can provide insights into their performance characteristics. The Table 4 below shows the comparison highlights of proposed model with other DL models.

Through qualitative and quantitative analyses, the exhibited results substantiate the superior effec-tiveness of the suggested SODU2-NET model over the U-Net architecture in the domain of image saliency detection, and with other popular DL-based image saliency detection algorithms, proving that the suggested approach can extract saliency maps with greater accuracy.

DeepLab stands as a widely recognized semantic segmentation model, distinguished by its application of dilated convolutions for capturing multi-scale insights. Notably, it integrates an ASPP module, making it easier to extract features across varying dilation rates to enhance contextual comprehension. While DeepLab achieves commendable accuracy in image segmentation, it's worth noting that its utilization might entail a greater demand for computational resources compared to the U-Net architecture.

FCN was one of the pioneering models for end-to-end pixel-wise segmentation. It does not have skip connections like U-Net, This limitation can restrict its capacity to capture

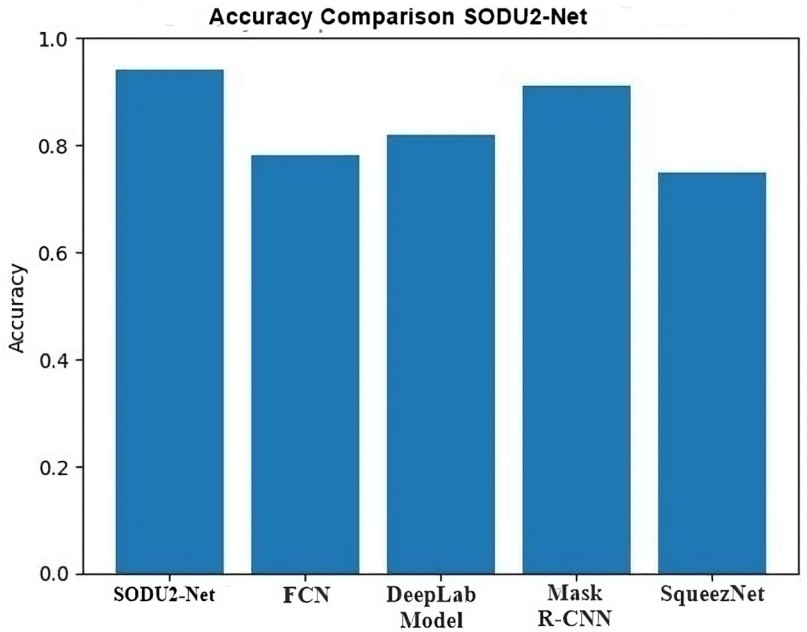

**Figure 18 Accuracy comparison in salient object detection.**

**Table 4 Comparative analysis.**

| Model | Key feature | Accuracy |
|---|---|---|
| Proposed SODU2-NET | It performs well in preserving fine details and capturing object boundaries. | 0.94 |
| FCN | It may struggle with preserving object boundaries and handling small or detailed salient objects | 0.78 |
| DeepLab | DeepLab may have limitations in preserving fine details and object boundaries compared to U-Net. | 0.82 |
| Mask R-CNN | Mask R-CNN offers object-level segmentation, but it may require additional post-processing steps for salient object detection. | 0.91 |
| SqueezeNet | It may sacrifice some accuracy compared to more complex models. | 0.75 |

intricate nuances and delineations within saliency maps. FCN may not perform as well as U-Net in tasks with smaller objects or complex scenes.

Mask R-CNN is well-suited for instance-level segmentation tasks, where precise segmentation of multiple object instances is required, but it can be computationally intensive.

SqueezeNet is a lightweight model suitable for classification tasks in resource-constrained environments, but it may not achieve the same level of accuracy as U-Net or mask R-CNN in complex segmentation tasks.

## CONCLUSION

Saliency object detection plays an important role in recognizing an important object from images or videos. The model consists of full feature fusion, residual blocks, squeeze and excitation, atrous spatial pyramid pooling, and U-Net architecture. The proposed SODU2-NET model follows the standard architecture of the U-Net by adding residual blocks and

ASPP modules for better feature extraction. The decoder part contains the attention module that focuses on the salient object of the image. The introduction of this unique dataset serves the purpose of showcasing results with real-world images from the city of Changsha, China. This dataset, which has never been utilized before, adds an element of novelty to the study. There are multiple objects captured in this city for detecting salient objects. The model performs well on this data set and produces accurate feature maps of salient objects. Several datasets are also utilized to assess the model's effectiveness. The outcomes of the presented model SODU2-NET achieve higher results on other datasets as well.

## Future work

Future work could entail further refinement of the SODU2-NET model's architecture to push the bound-aries of saliency detection accuracy. This could involve exploring advanced attention mechanisms to enhance the model's capability to identify salient objects with greater precision and robustness. Additionally, investigating the integration of weakly supervised learning techniques and domain adaptation approaches could facilitate the model's generalization across diverse datasets and real-world scenarios. By delving deeper into these areas, we aim to improve the model's performance and applicability in various practical applications, ultimately advancing the field of saliency object detection.

### Funding

This research was funded by the EIAS Data Science & Blockchain Lab, Prince Sultan University. The Prince Sultan University paid the APC of this article. The funders had no role in study design, data collection and analysis, decision to publish, or preparation of the manuscript.

### Grant Disclosures

The following grant information was disclosed by the authors:
EIAS Data Science & Blockchain Lab, Prince Sultan University.
The Prince Sultan University.

### Competing Interests

The authors declare that they have no competing interests.

### Author Contributions

- Hyder Abbas conceived and designed the experiments, performed the experiments, performed the computation work, authored or reviewed drafts of the article, and approved the final draft.
- Shen Bing Ren performed the experiments, prepared figures and/or tables, and approved the final draft.

- Muhammad Asim analyzed the data, prepared figures and/or tables, and approved the final draft.
- Syeda Iqra Hassan conceived and designed the experiments, performed the experiments, performed the computation work, authored or reviewed drafts of the article, and approved the final draft.
- Ahmed A. Abd El-Latif analyzed the data, authored or reviewed drafts of the article, and approved the final draft.

## Data Availability

The data are available at Figshare: Abbas, Hyder (2025). Salient Object Detection Dataset. figshare. Figure. https://doi.org/10.6084/m9.figshare.24818454.v1.

Abbas, Hyder (2025). Raw Data. figshare. Dataset. https://doi.org/10.6084/m9.figshare.25824403.v1.

## Supplemental Information

Supplemental information for this article can be found online at http://dx.doi.org/10.7717/peerj-cs.2623#supplemental-information.

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
