# Peer review of "SODU2-NET: a novel deep learning-based approach for salient object detection utilizing U-NET"

_PeerJ Computer Science, doi:10.7717/peerj-cs.2623_

## Round 0.1 · original submission · Major Revisions

· Academic Editor

Major Revisions

We have received comments and suggestions from the reviewers. They suggest a major revision to the article. You are required to revise your manuscript by considering all the suggestions and comments. The revised manuscript will be subjected to a 2nd round review. Good luck.

**Language Note:** The review process has identified that the English language must be improved. PeerJ can provide language editing services - please contact us at [email protected] for pricing (be sure to provide your manuscript number and title). Alternatively, you should make your own arrangements to improve the language quality and provide details in your response letter. – PeerJ Staff

Reviewer 1 ·

Basic reporting

The paper seems to be well-written.
However, the formatting of the paper is not in the journal format.

The paper also requires thorough proofreading from some professional sources.
Recent paper published in 2023 in not available in literature survey. Authors must include some of them.

There must be a separate sub-section as "Our Contributions" in the Introduction section. Also, it should highlight 4-5 main contributions regarding novelty.

Code snippets or screenshots should be placed in a GitHub repository along with the implementation code.

Experimental design

Several datasets are used in this case.
However, the authors are required to use some more real time data sets for the validation of the model proposed in this study.

Figure 4 is not at par with the publication guidelines. It is expected to fix the figure dimensions and blur effect.

Compare your technique with recent state-of-the-art works/models.

Discussed data used in this work in detail as a separate sub-section.

Validity of the findings

The results are quiet appreciable from the experiments performed by the authors.
However, there is a strong scope for the ablation analysis for the present study.
The authors must integrate it with the findings.


Future directions of research as a separate sub-section should be mentioned.

Additional comments

The manuscript acknowledges that:
The authors would like to acknowledge the support of Prince Sultan University for paying the Article Processing Charges (APC) of this publication

However, in the submission, it is mentioned that the manuscript is not receiving any funding.
It is recommended that the authors mention the proper acknowledgement in the article and the system.

Reviewer 2 ·

Basic reporting

This work proposes an DL-based model SODU2-NET for automated saliency detection using 92 residual blocks, and Atrous Spatial Pyramid Pooling (ASPP). It seems that the main contribution of this work lies in modified U-NET that adds the residual blocks inside the encoder architecture. Here, authors use multiple metrics to identify and compare the issue of SOD in old 107 models. However, the concept of the U-NET used for salient object detection in this paper have been used in many existing works, and the incorporation of these well-developed methods into the proposed work cannot be contribution to salient object detection. In addition, some suggestions for authors are as follows:
1. The introduction and related works must provide a critical evaluation of the models used for SOD in previous studies. The main contributions of this article and the differences with existing methods should be made clear in the Introduction and Related Work sections.
2. Authors are advised to present a critical discussion, not just a descriptive summary of the topic in literature review and other subsections.
3. Some of the figures are used from other literatures, which violating copy right rules. It would be better to re-create the figures and cite them properly.
4. Furthermore, the references should be arranged in the order of their appearance.

The contribution of the paper is not adequate. The whole paper needs significant improvement. The content of the paper is ambiguous. Introduction is not well written, and authors are unable to justify their contributions. Moreover, the results of the proposed work are not clear.

Experimental design

Experimental work is not enough to show advantages of the proposed work over existing techniques. It would better to discuss technical aspects of the proposed work. For example, What are the reasons that proposed work is giving better mean average precision, f1-score etc.?

Validity of the findings

the contribution of the paper is not adequate and it is not properly verified.

---

## Round 0.2 · Minor Revisions

· Academic Editor

Minor Revisions

One of the reviewers is not convinced with the revised manuscript and pointed out some changes which were not addressed. You are required to revise and resubmit it.

Reviewer 1 ·

Basic reporting

The authors have significantly modified and updated the version of the manuscript.
However, several items need to be addressed as per the additional comments below:

Experimental design

The experimental design seems acceptable in the present format for the paper.

Validity of the findings

The authors have successfully provided sufficient results and outcomes related to the validity of the findings.
The validity is now acceptable in its present format.

Additional comments

The manuscript still needs some proofreading revisions and grammar-related updates.
The authors have not convincingly provided a proper, proofreadable version of the manuscript.
The authors are advised to complete and thoroughly proofread the manuscript and submit the tracked version in proper format.

---

## Round 0.3 · accepted · Accept

· Academic Editor

Accept

Authors have addressed all the comments raised by reviewers. The paper may be accepted in the current form.